# Signal Enhancement in Magnetoelastic Ribbons Through Thermal Annealing: Evaluation of Magnetic Signal Output in Different Metglas Materials

**DOI:** 10.3390/s25123722

**Published:** 2025-06-13

**Authors:** Georgios Samourgkanidis, Dimitris Kouzoudis, Panagiotis Charalampous, Eyad Adnan

**Affiliations:** 1Department of Civil and Environmental Engineering, University of Cyprus, Nicosia 1678, Cyprus; 2Department of Chemical Engineering, University of Patras, 26504 Patras, Greece; kouzoudi@upatras.gr (D.K.); charalampous@upatras.gr (P.C.); 3Faculty of Engineering Technology and Science, Higher Colleges of Technology, Al Ain 64141, United Arab Emirates; eadnan@hct.ac.ae

**Keywords:** magnetoelastic materials, Metglas ribbons, thermal treatment optimization, signal enhancement, sensing applications

## Abstract

This study explores the impact of thermal annealing on the magnetic signal enhancement of three distinct Metglas ribbon materials: 2826MB3, 2605SA1, and 2714A. Each material underwent a systematic annealing process under a range of temperatures (50–500 °C) and durations (10–60 min) to evaluate the influence of thermal treatment on their magnetic signal response. The experimental setup applied a constant excitation frequency of 20 kHz, allowing for direct comparison under identical measurement conditions. The results show that while all three alloys benefit from annealing, their responses differ in magnitude, stability, and sensitivity. The 2826MB3 and 2605SA1 ribbons exhibited similar enhancement patterns, with maximum normalized voltage increases of 75.8% and approximately 70%, respectively. However, 2605SA1 displayed a more abrupt signal drop at elevated temperatures, suggesting reduced thermal stability. In contrast, 2714A reached the highest enhancement at 86.8% but also demonstrated extreme sensitivity to over-annealing, losing its magnetic response rapidly at higher temperatures. The findings highlight the critical role of carefully optimized annealing parameters in maximizing sensor performance and offer practical guidance for the development of advanced magnetoelastic sensing systems.

## 1. Introduction

Sensors play a pivotal role in modern infrastructure, acting as the backbone of intelligent monitoring systems across a wide range of industries. They enable the continuous, real-time tracking of critical physical and chemical parameters, such as temperature, pressure, humidity, pH levels, and pollutant concentrations, which are essential for the efficient functioning and long-term sustainability of contemporary societies. In the context of urban development, sensors form the foundation of smart city technologies by regulating traffic flow, optimizing energy consumption, and managing public safety systems [1]. Environmental monitoring applications rely heavily on sensors to assess air quality by detecting gases like ${\mathrm{CO}}_{x}$, ${\mathrm{NO}}_{x}$, and particulate matter, as well as to track water quality parameters such as turbidity, pH, and the presence of harmful contaminants [2,3]. Moreover, advanced sensor networks are employed in waste water and sewer systems to detect anomalies, prevent blockages, and reduce pollution, thereby contributing to ecological protection and resource management [4]. In agriculture, sensor technologies are integrated into precision farming systems to monitor soil moisture, nutrient content, and crop health, ultimately leading to improved yield, reduced resource consumption, and more sustainable food production practices [5,6].

Among the diverse range of materials utilized in sensor technologies, magnetoelastic alloys hold a particularly valuable position due to their unique combination of magnetomechanical coupling, affordability, and functional robustness [7,8,9,10]. These materials exhibit the distinct ability to undergo elastic deformation in the presence of an external magnetic field, and conversely, to develop magnetization when subjected to mechanical strain [11,12]. This bidirectional interaction underpins their functionality in various applications, most notably in anti-theft security systems, where they are commonly embedded as thin tags in retail products. Magnetoelastic alloys are typically composed of amorphous metallic structures, more widely known as metallic glasses or Metglas, which are formed by rapidly cooling molten metal to prevent crystallization [13,14,15]. Their non-crystalline nature imparts high electrical resistivity and significantly reduces eddy current losses, making them well suited for use in high-efficiency transformers [16,17,18] and low-loss magnetic sensors [19,20]. Additionally, their soft magnetic properties, characterized by a near-linear relationship between the applied magnetic field and the resulting magnetization [21,22,23], enable predictable and sensitive magnetic responses. This makes them particularly advantageous for use in dynamic sensing environments where signal accuracy and responsiveness are critical. In practical applications, magnetoelastic ribbons have been effectively utilized in systems such as magnetoelectric sensors [24,25], gas sensors [26,27,28], bio-sensors [29,30], and vibration detectors [31,32,33,34,35]. These devices rely on the material’s sensitivity to mechanical or environmental changes, enabling them to function as highly responsive transducers. The operating principle of magnetoelastic sensors is analogous to that of piezoelectric crystals in the magnetic domain, where fine-tuning the magnetic response is essential for accurately capturing even subtle magnetic field variations.

In this study, three different Metglas alloys, 2826MB3, 2605SA1, and 2714A, were selected based on their distinct magnetic and structural properties that make them suitable for sensing applications. Metglas 2826MB3 is an iron–nickel-based alloy offering medium saturation induction, low magnetostriction, and high corrosion resistance. It is commonly used in field sensors [20], magnetomechanical systems [36,37], and high-frequency magnetic cores [38], with the flexibility to be annealed for high permeability or a tailored BH loop shape, making it highly adaptable to different magnetic environments. Metglas 2605SA1, an iron-based alloy, is engineered for applications requiring extremely low core losses and high permeability. Its excellent performance in distribution transformers [39], current sensors [40], and motors [41] is attributed to its ability to minimize energy dissipation, particularly at both low and high frequencies. Lastly, Metglas 2714A is a cobalt-based alloy notable for its near-zero magnetostriction and ultrahigh magnetic permeability, making it ideal for ultra-sensitive applications such as magnetic amplifiers [42], shielding [43], and precision sensors [44]. It combines extremely low core loss with a high squareness ratio and can be annealed for a linear BH response.

Several prior studies have investigated the effects of annealing on various Metglas alloys, revealing how thermal treatment can significantly alter their magnetic properties. Cadogan et al. [45] investigated the embrittlement behavior of Metglas 2605S2, identifying a ductile-to-brittle transition near 310 °C, marked by a five-fold reduction in fracture toughness. Using Mössbauer spectroscopy, they observed increased magnetic texturing in both the bulk and surface of the ribbons upon annealing, linked to stress relaxation. Despite these changes, no significant structural modifications were detected via X-ray diffraction, emphasizing the subtle yet critical effects of thermal treatment on the alloy’s mechanical and magnetic properties. Deng et al. [46] found that annealing FeBSiC Metglas at 360 °C for 10 minutes produces a mixed amorphous crystalline phase, leading to improved piezomagnetic properties and reduced magnetic loss. This optimized structure enhanced ME coupling in composite sensors, increasing Q by up to 37.5% and lowering the equivalent magnetic noise, thereby significantly boosting low-frequency detection sensitivity. Lee et al. [47] reported that annealing the Co-based amorphous alloy ${\mathrm{Co}}_{67}{\mathrm{Fe}}_{3}{\mathrm{Cr}}_{3}B_{12}{\mathrm{Si}}_{15}$ at 350° for 1 hour significantly enhanced its magnetic properties, making it suitable for use in low-noise flux-gate magnetometers. The optimized ribbons demonstrated a noise level of 2.4 nT at 1 Hz and 0.1 nT peak-to-peak in a 1 Hz bandwidth. These findings highlight the alloy’s potential for high-frequency sensing applications and its suitability as a long-term, stable core material in precision magnetic devices. Sun et al. [48] compared the magnetomechanical performance of Fe-based amorphous ribbons, Vitrovac 7600 (FeCoSiB) and Metglas 2605SA1 (FeSiB), under various annealing conditions. They found that Metglas 2605SA1 significantly outperformed Vitrovac, achieving a maximum coupling factor of 73% and a magnetomechanical efficiency factor ($k^{2}$Q) as high as 16 after partial crystallization at elevated annealing temperatures. In contrast, Vitrovac exhibited a peak coupling factor of 23% and lower overall efficiency. These results confirm that appropriately annealed Metglas 2605SA1 ribbons offer superior power conversion efficiency, making them better suited for applications such as magnetostrictive actuators and acoustically driven antennas. Wang et al. [49] developed a novel multi-component Fe-based amorphous alloy (FeMnCuMoCPSiB) featuring a surface crystallization layer composed of ultra-fine nano α-Fe grains. This engineered structure resulted in superior soft magnetic properties (SMPs), including a high saturation magnetic induction ($B_{s}$) of 1.67 T, low coercivity ($H_{c}$) of 1.6 A/m, and high initial permeability ($\mu_{i}$) of 0.00093 at 1 kHz, significantly outperforming commercial Metglas 2605. Their findings underscore the role of nanoscale surface precipitation in enhancing, rather than degrading, magnetic performance, and present an effective pathway for advancing soft magnetic material design for applications in electronics and energy systems. Palneedi et al. [50] demonstrated a novel fabrication method for magnetoelectric (ME) heterostructured films by depositing PZT thick films onto Metglas substrates using aerosol deposition, followed by rapid annealing via intense pulsed light (IPL). This approach enabled effective crystallization of the piezoelectric phase without thermally degrading the Metglas, overcoming a major limitation in traditional high-temperature processing. Through optimization of IPL pulse duration, particularly at 0.75 ms, they achieved significantly enhanced ME coupling, reaching around 20 ${\mathrm{Vcm}}^{-1}{\mathrm{Oe}}^{-1}$, which is an order of magnitude higher than previously reported values. Their work highlights the potential of IPL-treated PZT/Metglas composites for high-performance, miniaturized ME devices such as sensors, energy harvesters, and RF components.

Building upon our previous work [51], which focused exclusively on the annealing behavior of Metglas 2826MB3 at a fixed excitation frequency of 1 kHz, the present study extends the investigation in two significant directions. First, it broadens the scope by comparing the thermal response of three distinct Metglas alloys, 2826MB3, 2605SA1, and 2714A, under identical annealing conditions, allowing for a direct evaluation of compositional effects on magnetoelastic signal enhancement. Second, this work advances the frequency domain of analysis by shifting to a higher excitation frequency of 20 kHz, providing new insights into the performance of these materials in more demanding sensing environments. Through this comparative and frequency-expanded approach, this study aims to offer a more comprehensive understanding of how thermal treatment influences magnetic signal output across different Metglas compositions.

## 2. Materials and Methods

In this study, three different magnetoelastic materials were selected for investigation, Metglas 2826MB3, Metglas 2605SA1, and Metglas 2714A, manufactured by Metglas Inc. (Conway, SC, USA) [52]. Figure 1 presents the hysteresis loops for all three materials, while Table 1 summarizes key general properties as provided by the manufacturer. Among the three Metglas types, 2605SA1 exhibits the highest value of saturation magnetic induction, whereas 2714A has the lowest. Regarding crystallization temperature, 2714A shows the highest value, reaching up to 550 °C, also based on manufacturer data. This temperature is particularly significant, as it marks the threshold at which the alloy transitions from its amorphous state, leading to immediate and pronounced changes in its magnetic properties, an effect that will be discussed further in Section 3.

Each material was obtained in ribbon form with a uniform width of 1/4 inch and a thickness of 30μm, and all ribbons were cut to a consistent length of 2.5 cm to ensure uniformity during experimentation; this length was selected because it corresponds to half the length of the 5 cm detection coil, so as to ensure the ribbon is fully inside the detection coil. For each material type, one ribbon was retained in its unannealed condition to serve as a control specimen. The remaining ribbons underwent thermal annealing under various conditions to evaluate the effect of heat treatment on the magnetic signal response. The annealing temperature was varied from 50 °C to 500 °C in increments of 50 °C, and the annealing time ranged from 10 to 60 min in 10 min intervals. This matrix of temperature and time resulted in 60 unique annealing conditions per material type. Consequently, 60 ribbons were prepared and thermally treated for each Metglas alloy, leading to a total of 180 ribbon specimens analyzed in this study (Figure 2a–c).

The annealing process was conducted identically for all three Metglas materials. For each target temperature, the furnace (Figure 2d) was first allowed to stabilize at the setpoint. A group of six ribbons from the same Metglas type was then placed into the furnace, with each ribbon assigned a specific annealing time ranging from 10 to 60 min in 10 min increments. After each time interval, the corresponding ribbon was individually removed and allowed to cool naturally in ambient conditions. This process was repeated across all selected temperature levels and for each alloy type. To ensure more uniform thermal exposure and controlled cooling, each ribbon was placed between two thick blocks of aluminum alloy 7075 before being inserted into the furnace. This setup served two key purposes. First, the aluminum blocks helped establish a stable thermal environment around the ribbon, as their higher thermal mass minimized temperature fluctuations that could arise if the ribbon was freely exposed in the furnace. Second, removing the ribbon while still sandwiched between the preheated blocks allowed it to cool more gradually, reducing the risk of sudden temperature drops due to its extremely thin geometry. This method was chosen to improve consistency in the annealing process and minimize unwanted thermal stress effects that could impact the magnetic signal performance of the materials.

The experimental setup used to evaluate the magnetoelastic signal response of each ribbon specimen is illustrated in Figure 3. The system consists of two Helmholtz coil pairs: the inner coils are connected to a DC power supply that generates a static bias magnetic field, while the outer coils are driven by a function generator to produce a low-amplitude alternating magnetic field. Positioned at the center of these coils is a detection coil, into which the magnetoelastic ribbon is inserted. This detection coil captures the time-varying magnetic signal induced by the ribbon’s ferromagnetic behavior. The output signal is then amplified using a pre-amplifier and its amplitude measured with a digital voltmeter. The test procedure is as follows: a ribbon is placed inside the detection coil, and an AC magnetic field is applied in the form of a sinusoidal excitation, defined as ΔH = $H_{0}$cos(2πft) with a fixed excitation frequency of 20 kHz. Simultaneously, a constant DC magnetic field $H_{\mathrm{DC}}$ is applied to bias the ribbon to its optimal working point. This biasing condition is standard in magnetoelastic sensing applications, as it maximizes the linearity of the sensor’s input-output response and improves signal clarity. As a result of its ferromagnetic nature, the ribbon develops a time-varying magnetization ΔM = $M_{0}$cos(2πft), which induces a changing magnetic flux in its surroundings. According to Faraday’s law of electromagnetic induction, this changing flux generates an AC voltage across the detection coil V = $V_{0}$sin(2πft). This voltage signal, which is amplified and read by the voltmeter, is directly related to the rate of change of magnetization. Given the relationship χ = ΔM/ΔH and since both f and $H_{0}$ are held constant throughout the experiments, any observed increase in voltage amplitude $V_{0}$ can be attributed to an increase in magnetic susceptibility χ of the ribbon. This method was uniformly applied across all three Metglas materials under study, allowing for a direct comparison of their magnetoelastic signal responses under identical excitation and biasing conditions [51].

The thermal annealing process serves to relieve internal stresses within the material, which are a primary contributor to elevated magnetic anisotropy energy and, consequently, a high anisotropy field [53]. By reducing this internal stress, the anisotropy field is lowered, resulting in an increase in magnetic susceptibility. In this study, the method involves comparing the measured voltage signals of each ribbon before and after annealing to identify the optimal annealing conditions that maximize magnetic response.

At this point, we would like to provide a more detailed explanation of why 20 kHz was chosen as the fixed excitation frequency for this study. This decision was based on insights from two previous studies conducted by our research group. In the first study [37], Metglas 2826MB3 demonstrated a broad and stable magnetic signal response across a frequency range of 300 Hz to 50 kHz, confirming its consistent performance throughout. In a follow-up study [51], we specifically examined the annealing behavior of 2826MB3 at 1 kHz, investigating how thermal treatment affected its signal output. With these results in mind, we chose 20 kHz in the present work for two key reasons. First, we wanted to explore how 2826MB3 performs at a higher frequency within its known stable range, allowing for direct comparison with its behavior at 1 kHz. Second, by including additional materials, 2605SA1 and 2714A, in the testing process at the same frequency, we aimed to investigate their magnetic signal responses under similar high-frequency conditions. This not only allows for a consistent comparison across different alloys in the current study but also establishes a solid reference point for future investigations where these materials will be examined over a broader range of excitation frequencies.

## 3. Results and Discussion

The sensing procedure was implemented as follows: Initially, with the detection coil empty (Figure 3), the amplifier was calibrated so that the voltmeter displayed a baseline voltage of 1 mV, which served as the system’s reference signal. After calibration, an unannealed ribbon was placed inside the detection coil, and the resulting output voltage was recorded as $V_{a}$. Following the annealing process, each treated ribbon was similarly inserted into the coil, and the new voltage output, $V_{b}$, was measured. The difference, defined as ΔV = $V_{b}$− $V_{a}$ was used to assess the effect of annealing on the ribbon’s signal response. All measurements were conducted under identical experimental conditions, with a DC bias field applied at its optimized level to enhance the magnetic response and improve signal detection sensitivity.

### 3.1. Metglas 2826MB3

Figure 4a presents a 2D color map illustrating the normalized ΔV (%) response of the 2826MB3 ribbons as a function of annealing temperature (vertical axis) and time (horizontal axis). The color scale to the right indicates the degree of signal enhancement, with values ranging from approximately 31% to 76%. Compared to our earlier study at 1 kHz [51], where a similar mapping was performed, the overall shape of the response remains consistent at 20 kHz. However, a noticeable difference lies in the magnitude of the enhancement: the previous results showed a peak near 100%, while the current measurements reach up to around 76%. According to Table 1, the Curie temperature of the alloy is approximately 353 °C and the onset of crystallization begins at around 410 °C. Notably, a significant rise in signal begins near 300 °C, transitioning from mid-range values (teal-green, around 45%) to much higher values (around 75%, red). The red contours, representing peak enhancement, are spread over a wide time window and gradually shift to lower temperatures as the annealing time increases. This suggests a temperature–time trade-off to achieve optimal signal amplification. At the upper-right region of the plot, a shift to dark blue (around 31%) indicates a drop in signal strength at prolonged annealing times and high temperatures (470–500 °C). This decline likely corresponds to structural changes in the material, possibly due to ongoing crystallization as the alloy is already 60 °C to 90 °C above its critical transformation temperature.

Figure 4b displays the 3D surface representation of ΔV, offering a clearer view of the signal enhancement region. The elevated area, corresponding to the highest normalized values, spans a range of approximately 70–75%. Surface fitting analysis identifies two distinct peak points: one at 457 °C and 22 min, and another at 356 °C and 41 min, both reaching a maximum value of 75.8%. Conversely, the minimum response of 32.3% is observed at the far end of the plot, where both temperature and time are at their highest.

Figure 5 presents the normalize ΔV variation over time for Metglas 2826MB3 annealed at three different temperatures. These temperatures were selected symmetrically within the investigated range to more clearly demonstrate the signal enhancement. As the annealing temperature increases to 250 °C, the output signal exhibits strong temporal stability, with the normalized ΔV remaining within a narrow range of 45–50% as the annealing time progresses. As the annealing temperature increases from 250 °C to 350 °C, the behavior of the normalized ΔV begins to change, showing a more gradual increase that suggests the presence of an optimal annealing duration. Specifically, ΔV increases steadily from 63% to 74% between 10 and 40 min of annealing, after which it saturates at around 73%. This trend indicates that the additional thermal energy promotes further mechanical stress relaxation, enhancing the output signal. Importantly, this enhancement occurs without significantly affecting the amorphous structure of the material, as crystallization remains absent within this temperature range. When the annealing temperature is further increased from 350 °C to 450 °C, the behavior of the normalized ΔV reverses compared to the trend observed at 350 °C. In this case, the material reaches its peak response after only 10 min of annealing, followed by a steady decline beginning around 25–30 min. This suggests that the higher thermal energy at 450 °C begins to affect the internal structure of the material, likely due to the onset of crystallization. This structural change negatively impacts the magnetic properties, leading to a reduction in signal performance. At this annealing temperature, the normalized ΔV decreases by 25%, dropping from 73% to 48%

### 3.2. Metglas 2605SA1

For the Metglas 2605SA1 alloy, the overall behavior closely resembles that of 2826MB3. As shown in the corresponding color map (Figure 6a), an optimal annealing region is evident at elevated temperatures and across a range of times, similar to the previous case. Additionally, a steep drop in signal is observed at the upper-right section of the map, indicating degradation at prolonged annealing times and high temperatures. However, this decline begins earlier than in 2826MB3, as the blue region extends toward shorter annealing durations. While the maximum signal enhancement in the red region remains comparable, at around 70%, a key difference lies in the severe drop in performance within the blue region, where the normalized ΔV falls to nearly 2%. This suggests a substantial loss of magnetic properties under certain conditions. According to Table 1, the Curie and crystallization temperatures for this alloy are 395 °C and 510 °C, respectively, both higher than those of 2826MB3. This explains the sharp decline in magnetic performance observed near 500 °C, as seen in the color map. In contrast, for 2826MB3, despite its lower crystallization temperature of 410 °C, the material maintains a strong signal response even at elevated temperatures and short annealing times, suggesting a lower crystallization rate compared to 2605SA1. Another notable feature is the presence of orange to light red zones at lower temperatures (below 200 °C), indicating a signal enhancement slightly exceeding 50%.

The 3D surface representation of ΔV for this alloy exhibits a more uneven profile (Figure 6b), with pronounced peaks and valleys across the temperature and time axes. The decline in signal at higher temperatures is significantly steeper compared to the previous alloy, with the fitted minimum value dropping to just 1.6% at 500 °C and 60 min. In contrast, a single distinct peak is observed at 412 °C and 31 min, marking the optimal annealing conditions for this material.

Figure 7 illustrates the normalized ΔV variation over time for this alloy, annealed at three different temperatures. Compared to Metglas 2826MB3, the behavior appears more unstable. At 250 °C, a sharp decrease of nearly 20% in ΔV is observed between 30 and 40 min of annealing, indicating higher signal variability within this temperature range, an effect that is also clearly visible in the 3D plot shown in Figure 6b. In contrast, annealing at 350 °C results in more consistent behavior, with ΔV gradually increasing from 43% to 58% over time, without any sign of saturation, suggesting that optimal conditions for further annealing may exist. A similar trend is seen at 450 °C, where ΔV first increases slightly to around 65% at 30 min, and then drops sharply to approximately 23% at 60 min. Unlike the behavior observed in 2826MB3, this abrupt decline suggests a higher rate of structural transition, due to crystallization, indicating a greater sensitivity of this alloy to elevated annealing temperatures.

### 3.3. Metglas 2714A

This ribbon displays behavior markedly different from the other alloys (Figure 8a). As shown in its color map, the annealing process maintains relatively stable signal enhancement over a broad range, with the green region averaging around 52%. The area of highest enhancement (red) appears in the upper-left portion of the map, while the blue zone, indicating signal suppression, is widespread across much of the high-temperature range (>400 °C). According to Table 1, this alloy’s crystallization temperature is 550 °C. The extensive presence of the blue region, combined with low signal values similar to those observed in 2605SA1, suggests that temperatures exceeding approximately 72% of the crystallization point significantly degrade the material’s performance, highlighting its sensitivity to thermal treatment.

The 3D surface plot of ΔV for this alloy clearly reveals a temperature boundary (Figure 8b), with a sharp decline in signal occurring beyond 400 °C. The peak ΔV value, which is the highest among all tested ribbons at 86.8%, is observed near 439 °C and 10 min. However, surface fitting analysis refines this to a maximum at approximately 9 min and 442 °C. This further underscores the alloy’s sensitivity to thermal treatment, as even slight variations in time and temperature significantly impact performance.

Figure 9 shows ΔV as a function of time for Metglas 2714A annealed at three different temperatures. What sets this alloy apart from the other two is its pronounced response at the highest annealing temperature (450 °C). The data clearly indicate that the thermal energy rapidly affects the crystallization structure, as evidenced by a sharp decline in the initially strong signal enhancement occurring between 10 and 20 min of annealing. The magnetic signal drops sharply from 86% to below 10% and then stabilizes within that range. In contrast, at the other two annealing temperatures, the behavior is considerably more stable, especially when compared to 2605SA1. At 250 °C, a slight decline is observed, but the signal remains relatively stable, while at 350 °C, a steady enhancement is seen over time.

### 3.4. Comparison Results

This paragraph compares and presents the optimum results for each Metglas material. Figure 10 illustrates the ΔV behavior over time at the optimum annealing temperatures, as determined through fitting analysis, for all three alloys. The results show that the 2826MB3 and 2605SA1 alloys exhibit similar behavior at their respective optimum annealing temperatures, differing primarily in their annealing durations. Specifically, 2826MB3 reaches its maximum response at around 20 min of annealing, followed by a steady decline, while 2605SA1 peaks at around 30 min and then also declines similarly. As shown, the optimum annealing conditions for Metglas 2826MB3 are 457 °C for 22 min, and for Metglas 2605SA1, 412 °C for 31 min. In contrast, the 2714A alloy displays significantly different behavior. As discussed earlier, at its optimum annealing temperature, it shows a sharp decline in signal without a distinct peak within the tested annealing time range. The output drops by nearly 70% between 10 and 20 min of annealing and then stabilizes, with a more gradual decline over time. Nevertheless, this alloy achieves the highest magnetic signal enhancement, at 86.8%, at approximately 9 min of annealing at 442 °C, according to the fitting analysis, which surpasses the enhancements observed in the other two alloys.

### 3.5. Reported Benefits of Thermal Annealing in the Literature

Table 2 compiles reported improvements in magnetoelastic and related properties achieved through thermal annealing, drawing from both the prior literature and the present study. The data illustrate a consistent trend: carefully optimized annealing protocols lead to significant enhancements in performance metrics such as magnetic permeability, coupling efficiency, noise suppression, magnetic signal output, etc. Ref. [45] reported a marked improvement in the in-plane magnetic texture of Fe-based Metglas 2605S2 after annealing at 360 °C for 5 min, where the bulk intensity ratio increased from 2.2 to 2.8, attributed to stress-relief and structural relaxation effects. In [46], annealing Mn-PMNT/FeBSiC composites at 360 °C led to a 36.8% increase in the piezomagnetic coefficient, directly enhancing ME sensitivity at low frequencies. Ref. [47] observed significant noise reduction in flux-gate magnetometers by annealing ${\mathrm{Co}}_{67}{\mathrm{Fe}}_{3}{\mathrm{Cr}}_{3}B_{12}{\mathrm{Si}}_{15}$ at 350 °C for 60 min, achieving a noise level of 2.4 nT at 1 Hz, twice as quiet as sensors using untreated Metglas 2714A cores. Ref. [48] compared two amorphous alloys and found that annealing Metglas 2605SA1 at 440 °C for 20 min yielded a coupling factor of 73%, greatly surpassing Vitrovac 7600, which peaked at 23% at 390 °C for 20 min. In [49], a multi-component Fe-based alloy exhibited not only a 136.8% increase in initial permeability (from 0.0057 to 0.00135) after annealing at 523 °C but also a 50% reduction in core losses, emphasizing that thermal treatment can simultaneously optimize energy efficiency and magnetic performance. Meanwhile, Ref. [50] achieved a record-breaking ME coupling of 20 ${\mathrm{Vcm}}^{-1}{\mathrm{Oe}}^{-1}$ in PZT/Metglas composites using intense pulsed light (IPL) annealing, showcasing a novel method that enhances piezoelectric phase crystallization without compromising the magnetic substrate. In the present study, significant signal amplification was achieved through conventional thermal annealing: Metglas 2826MB3 showed a peak normalized signal gain of 75.8%, Metglas 2605SA1 reached 70%, and Metglas 2714A achieved the highest increase at 86.8%. These findings further affirm the critical role of carefully tuned annealing parameters in optimizing the performance of magnetoelastic and magnetoelectric systems across diverse applications.

## 4. Conclusions

The results of this study demonstrate that thermal annealing significantly influences the magnetoelastic signal response of Metglas ribbon materials, with each alloy exhibiting distinct behavior under identical processing and measurement conditions. The 2826MB3 alloy showed stable signal enhancement with a broad optimal annealing range and a peak normalized ΔV of 75.8%, confirming its suitability for thermal treatment. The 2605SA1 ribbons presented a similar trend in maximum enhancement (70%) but exhibited sharper degradation at elevated temperatures, indicating greater thermal sensitivity. In contrast, the 2714A alloy displayed a unique profile, achieving the highest recorded enhancement at 86.8% yet also experiencing a rapid decline in performance at temperatures exceeding 72% of its crystallization point. This highlights the alloy’s strong responsiveness but also its vulnerability to over-annealing. These findings underline the importance of the annealing conditions, specifically temperature and duration, to the specific thermal characteristics of each alloy. By identifying the optimal processing parameters, this work contributes to the advancement of magnetoelastic materials for sensing applications, offering a practical framework for enhancing signal output through controlled thermal treatment.

## Figures and Tables

**Figure 1 sensors-25-03722-f001:**
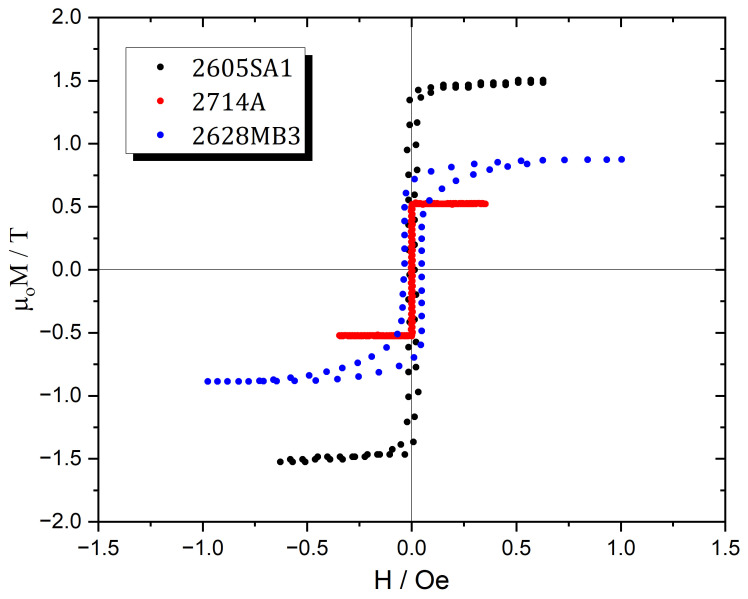
Magnetic hysteresis curves for all tested Metglas materials are presented. These curves correspond to the samples in their as-received state, prior to the annealing process.

**Figure 2 sensors-25-03722-f002:**
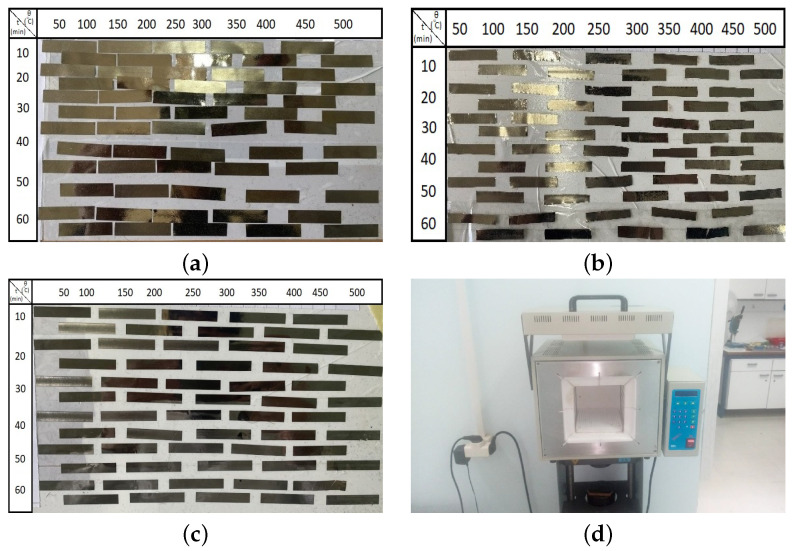
All ribbon specimens prepared for thermal annealing for each Metglas material: (**a**) Metglas 2826MB3, (**b**) Metglas 2605SA1, and (**c**) Metglas 2714A. (**d**) The furnace used for the annealing process.

**Figure 3 sensors-25-03722-f003:**
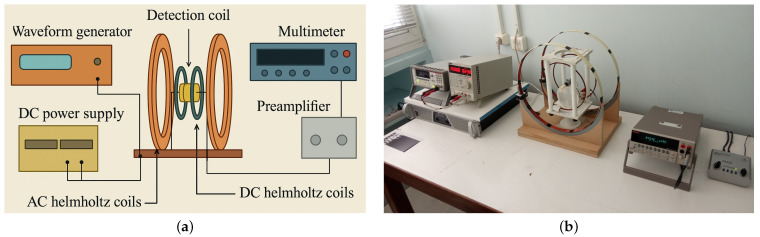
Experimental setup used for magnetoelastic signal measurement: (**a**) schematic diagram and (**b**) actual setup.

**Figure 4 sensors-25-03722-f004:**
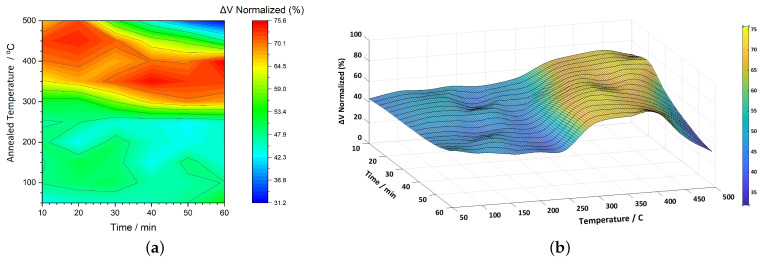
(**a**) a 2D color map and (**b**) a 3D surface plot of normalized ΔV (%) for 2826MB3 ribbons as a function of annealing temperature and time.

**Figure 5 sensors-25-03722-f005:**
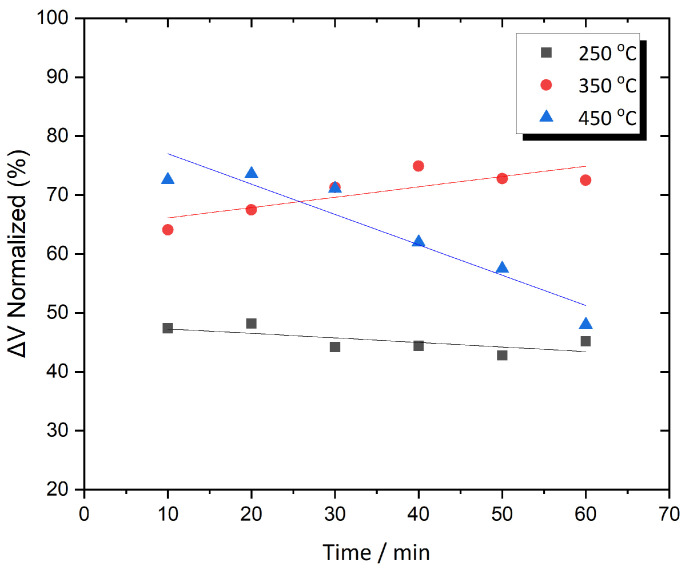
ΔV as a function of time for Metglas 2826MB3 annealed at three different temperatures.

**Figure 6 sensors-25-03722-f006:**
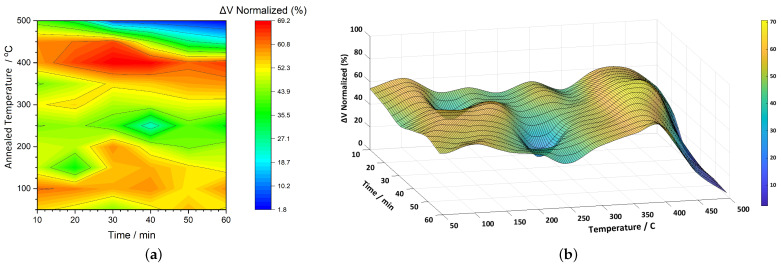
(**a**) a 2D color map and (**b**) a 3D surface plot of normalized ΔV (%) for 2605SA1 ribbons as a function of annealing temperature and time.

**Figure 7 sensors-25-03722-f007:**
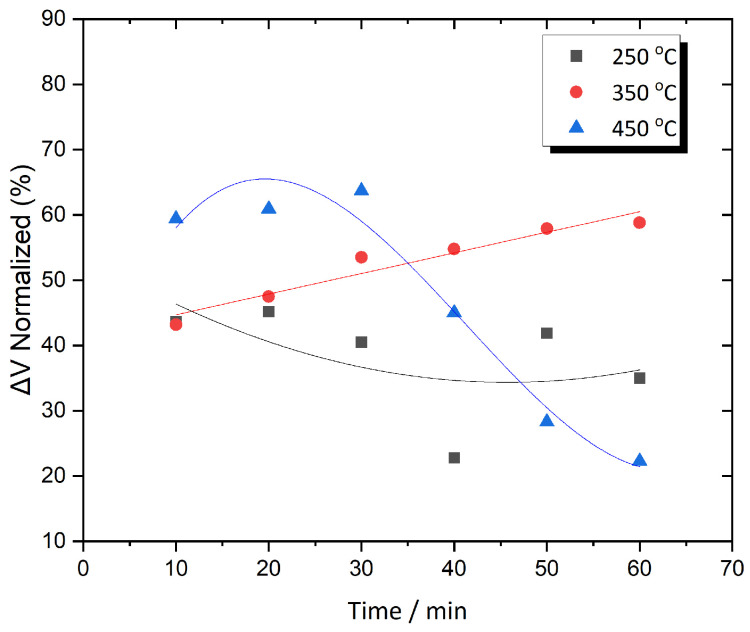
ΔV as a function of time for Metglas 2605SA1 annealed at three different temperatures.

**Figure 8 sensors-25-03722-f008:**
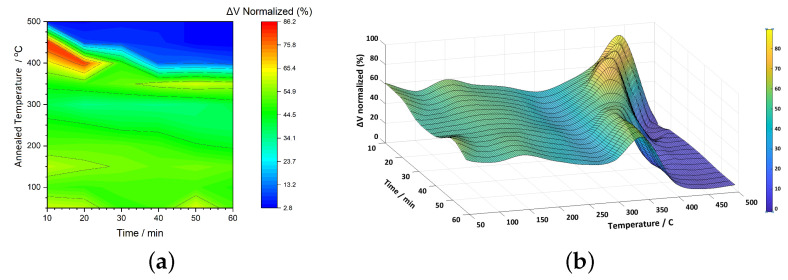
(**a**) a 2D color map and (**b**) a 3D surface plot of normalized ΔV (%) for 2714A ribbons as a function of annealing temperature and time.

**Figure 9 sensors-25-03722-f009:**
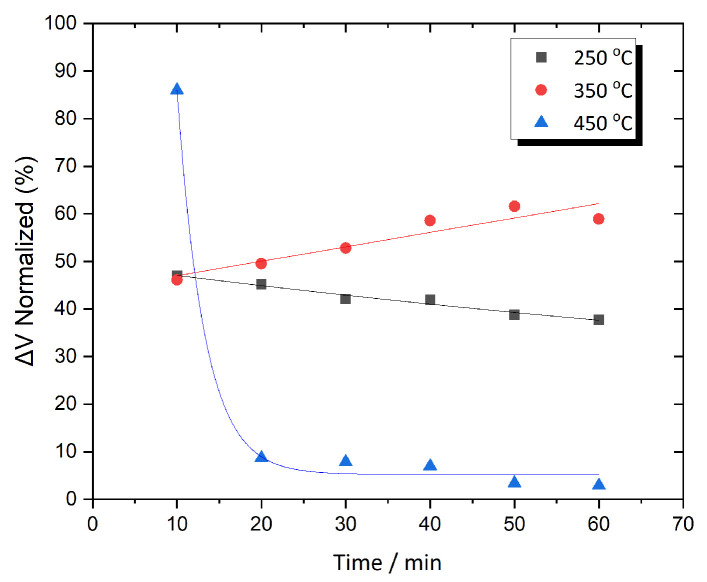
ΔV as a function of time for Metglas 2714A annealed at three different temperatures.

**Figure 10 sensors-25-03722-f010:**
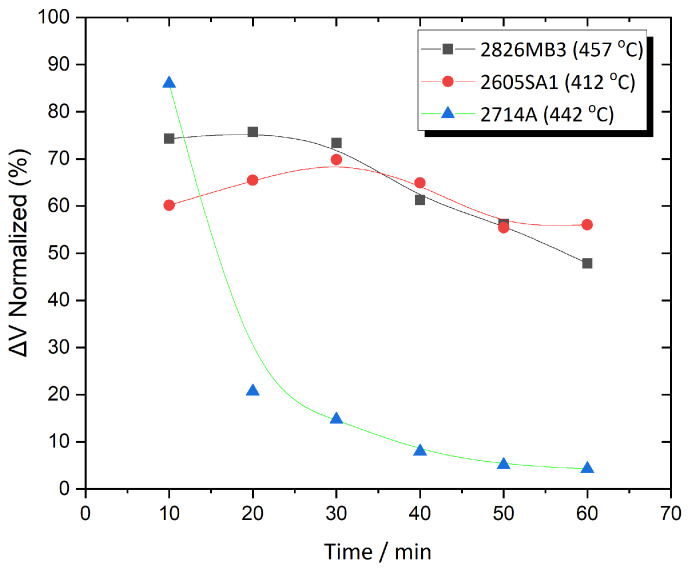
Comparison results: ΔV as a function of time for all alloys annealed at their respective optimum temperatures.

**Table 1 sensors-25-03722-t001:** Material properties of the Metglas ribbons, based on data from the Metglas Inc. (Conway, SC, USA) [52]. website [52].

Properties	2628MB3	2605SA1	2714A
Base Material	Fe-Ni	Fe	Co
Density (${\mathrm{g/cm}}^{3}$)	7.90	7.18	7.59
Saturation Induction (T)	0.88	1.56	0.57
Saturation Magnetostriction (ppm)	12	27	<0.5
Curie Temperature (°C)	353	395	225
Crystallization Temperature (°C)	410	510	550

**Table 2 sensors-25-03722-t002:** The literature and current work (CW) results are presented, where OAS refers to the optimal annealing settings used in each study.

Ref.	Alloy	OAS (°C-min)	Property	Improvement
[45]	${\mathrm{Fe}}_{78}{\mathrm{Si}}_{9}B_{13}$ (Metglas 2605S2)	360-5	In-plane magnetic texture	27.2%
[46]	Mn-PMNT/FeBSiC	360-5	Piezomagnetic coefficient	36.8%
[47]	${\mathrm{Co}}_{67}{\mathrm{Fe}}_{3}{\mathrm{Cr}}_{3}B_{12}{\mathrm{Si}}_{15}$	350-60	Noise level	108.3%
[48]	FeCoSiB (Vitrovac 7600)	390-20	Coupling factor	23%
	FeSiB (Metglas 2605SA1)	440-20	//	73%
[49]	${\mathrm{Fe}}_{74.89}{\mathrm{Cr}}_{3.04}{\mathrm{Ni}}_{1.23}{\mathrm{Co}}_{0.11}{\mathrm{Mn}}_{0.47}{\mathrm{Cu}}_{0.02}{\mathrm{Mo}}_{0.01}C_{0.11}P_{0.12}{\mathrm{Si}}_{9}B_{11}$	523-10	Magnetic permeability	136.8%
	${\mathrm{Fe}}_{78.32}{\mathrm{Mn}}_{0.49}{\mathrm{Cu}}_{0.02}{\mathrm{Mo}}_{0.01}C_{0.12}P_{0.13}{\mathrm{Si}}_{9.41}B_{11.5}$	521-10	Core losses	50%
[50]	Pb(Zr,Ti)$O_{3}$/FeSiB (PZT/Metglas 2605SA1)	300-2	Magnetoelectric coupling	≈1000%
CW	${\mathrm{Fe}}_{37}{\mathrm{Ni}}_{42}{\mathrm{Mo}}_{4}B_{17}$ (Metglas 2826MB3)	457-22	Magnetic signal	75.8%
	FeSiB (Metglas 2605SA1)	412-31	//	70%
	${\mathrm{Co}}_{66}{\mathrm{Fe}}_{4}{\mathrm{Ni}}_{1}B_{14}{\mathrm{Si}}_{15}$ (Metglas 2714A)	442-9	//	86.8%

## Data Availability

The data are available upon request from the authors.

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
