# Peer review of "Signal Enhancement in Magnetoelastic Ribbons Through Thermal Annealing: Evaluation of Magnetic Signal Output in Different Metglas Materials"

_sensors, 2025, doi:10.3390/s25123722_

Round 1

Reviewer 1 Report

Comments and Suggestions for Authors

Review report of sensors-3680173-v1

The manuscript describes the signal enhancement and evaluation of magnetic signal output in different Metglas ribbons: 826MB3, 2605SA1, and 2714A. The magnetoelastic property of the samples were enhanced through thermal treatment in a temperature-controlled furnace. The magnetic response of the annealed samples was investigated using homemade Helmholtz coil pairs and a digital voltmeter. This manuscript is partially organized and contains interesting details. However, the proposed idea and some results presented are an incremental improvement work of the previous report by the authors (reference #47). I would suggest revising the manuscript and including more information and data according to the comments below. The manuscript may be reconsidered after major revision.

  1. On page 4, line 142, the nominal composition of Metglas ribbons should be experimentally confirmed; for example, Energy Dispersive X-Ray Spectroscopy (EDX) technique. Then, the reported manuscript will be stronger.  
  2. On page 4, line 150, the experimental details of reported magnetic hysteresis must be provided for the readers. If magnetic hysteresis is taken from a reference, a specific citation must be given as well. More importantly, the magnetic hysteresis of the annealed samples is one of crucial factors describing the magnetic signal output.  
  3. On page 4, line 150, only one figure is presented, (a) is not needed.
  4. On page 4, line 150, Table1., a specific citation or a clear reference must be given.
  5. On page 5, lines 161-162, Figure 2., optical images of the different Metglas ribbons: 826MB3, 2605SA1, and 2714A should be presented with labeled dimensions.
  6. On page 6, line 190, a specific citation and a clear reference must be provided.
  7. On page 7, line 22, the vertical axis should be labelled as “Annealed temperature”.
  8. On pages 7-8, Figures 4 and 5 are presented with the same information. A line plot between annealed time vs altered voltage of some annealed temperatures will provide useful information. A comparison of magnetic signal output of the different excitation frequencies will be interesting to explore as well.
  9. On pages 9-10, Figures 6-7, and Figures 8-9 are presented with the same information as well.
  10. On page 10, line 285, the comparison of the magnetic signal output obtaining from annealed samples can be reported using a bar graph or line graph to summarize the results. The graph will assist the readers get a clear picture and understanding of the findings.

Author Response

Dear reviewer

Thank you very much for taking the time to review our manuscript and for providing insightful and constructive feedback to help improve its quality for publication. This document contains our detailed responses to each of your comments. A revised version of the manuscript, with all modifications clearly marked and underlined, is included in the accompanying PDF file.

Comments

Point 1: On page 4, line 142, the nominal composition of Metglas ribbons should be experimentally confirmed; for example, Energy Dispersive X-Ray Spectroscopy (EDX) technique. Then, the reported manuscript will be stronger.

  • Response 1: We sincerely thank the reviewer for this insightful and constructive suggestion. You are absolutely right that experimentally confirming the nominal composition of the Metglas ribbons using techniques such as Energy Dispersive X-ray Spectroscopy (EDX) would strengthen the manuscript. While the primary focus of our current study is on the sensing direction and magnetic signal behavior of the material, we fully acknowledge the importance of compositional verification. Unfortunately, due to limited access to EDX analysis facilities in our laboratory, we were unable to include new EDX measurements in this work. That said, in our previous collaborative research with colleagues from Spain, we conducted EDX analysis on similar Metglas ribbons (on 2826MB3 and 2714A). The results confirmed the nominal composition, and these findings are reflected in Table 2 of the current manuscript.

Point 2: On page 4, line 150, the experimental details of reported magnetic hysteresis must be provided for the readers. If magnetic hysteresis is taken from a reference, a specific citation must be given as well. More importantly, the magnetic hysteresis of the annealed samples is one of crucial factors describing the magnetic signal output.

  • Response 2: A reference has been added, and the caption of Figure 1 has been revised to incorporate the reviewer’s suggestion.

Point 3: On page 4, line 150, only one figure is presented, (a) is not needed.

  • Response 3: Corrected.

Point 4: On page 4, line 150, Table1., a specific citation or a clear reference must be given.

  • Response 4: A reference has been added to incorporate the reviewer’s suggestion.

Point 5: On page 5, lines 161-162, Figure 2., optical images of the different Metglas ribbons: 2826MB3, 2605SA1, and 2714A should be presented with labeled dimensions.

  • Response 5: Instead of modifying the image to include labels, we revised the corresponding paragraph to address the reviewer’s suggestion. This approach was chosen to avoid overlapping the ribbons in the image with additional graphics, which would obscure some of the ribbons corresponding to specific temperatures and times. However, if the reviewer insists on including labeled dimensions, we can provide a schematic of the ribbon with dimensions, either as part of Figure 5 or alongside the image of the furnace.

Point 6: On page 6, line 190, a specific citation and a clear reference must be provided.

  • Response 6: We would appreciate clarification regarding the reviewer’s request, as it is not entirely clear to us. Is the intention to include a reference for the detection method described? Nevertheless, we have included a reference to this method as presented in our previously published work.

Point 7: On page 7, line 22, the vertical axis should be labelled as “Annealed temperature”.

  • Response 7: Corrected to all figures.

Point 8: On pages 7-8, Figures 4 and 5 are presented with the same information. A line plot between annealed time vs altered voltage of some annealed temperatures will provide useful information. A comparison of magnetic signal output of the different excitation frequencies will be interesting to explore as well.

  • Response 8: Revised in accordance with the reviewer’s suggestion: Figures 4 and 5 have been combined, and an additional line plot illustrating the ΔV voltage versus time for selected temperatures has been included. The reviewer makes a valid point by suggesting an experiment to study the magnetic signal output as a function of excitation frequency. As our laboratory is focused on sensing applications of Metglas materials and this is an experiment we are already planning to include in our upcoming research.

Point 9: On pages 9-10, Figures 6-7, and Figures 8-9 are presented with the same information as well.

  • Response 9: The same approach as described in Response 8 has been applied here.

Point 10: On page 10, line 285, the comparison of the magnetic signal output obtaining from annealed samples can be reported using a bar graph or line graph to summarize the results. The graph will assist the readers get a clear picture and understanding of the findings.

  • Response 10: A bar and line graph has been added in accordance with the reviewer’s suggestion.

Reviewer 2 Report

Comments and Suggestions for Authors

The paper is an interesting experimental study on how thermal annealing affects ribbons made of three different Metglas materials (2826MB3, 2605SA1 and 2714A) with respect to their magnetic signal power. Ribbons were annealed at temperatures ranging from 50 to 500 oC in increments of 50 oC, whereas the annealing time for each case was from 10 to 60 minutes in 10 minute intervals. The magnetic signal power of each annealed (at specific temperature and time) ribbon was tested against a standard (untreated) counterpart, by vibrating both test specimens and measuring the magnetic signal enhancement at each test run. It results that ribbons of 2826MB3 present the most consistent behavior in terms of magnitude of magnetic signal enhancement due to the robustness of its performance throughout the range of annealing temperatures and times, as specified in the experiments.

The paper is well organized and written and results are stated in a quite clear manner. The introduction is quite long but well organized, which is a bonus: it provides the (not necessarily expert) reader with a clear description of the state-of-the-art on the magnetoelastic sensing principle and its applications.

A potential improvement would be to provide further comments on the selection of *one* vibrating frequency for the testing procedure (that of 20KHz, lines 198-207). If 2826MB3 has been found to exhibit consistent magnetic signal performance when tested between 300Hz and 50KHz (and has been specifically assessed at 1KHz by the past), one would have thought that annealed specimens of 2605SA1 and 2714A should also be examined at *more than one* frequencies inside the [300Hz-50KHz] range… This could also be an opportunity to test whether the annealing could considerably enhance the magnetic signal performance of such ribbons at  low vibration frequencies (as often experienced in mechanical applications)

Author Response

Dear reviewer

Thank you very much for taking the time to review our manuscript and for providing insightful and constructive feedback to help improve its quality for publication. This document contains our detailed responses to each of your comments. A revised version of the manuscript, with all modifications clearly marked and underlined, is included in the accompanying PDF file.

Comments

Point 1: A potential improvement would be to provide further comments on the selection of *one* vibrating frequency for the testing procedure (that of 20KHz, lines 198-207). If 2826MB3 has been found to exhibit consistent magnetic signal performance when tested between 300Hz and 50KHz (and has been specifically assessed at 1KHz by the past), one would have thought that annealed specimens of 2605SA1 and 2714A should also be examined at *more than one* frequencies inside the [300Hz-50KHz] range… This could also be an opportunity to test whether the annealing could considerably enhance the magnetic signal performance of such ribbons at  low vibration frequencies (as often experienced in mechanical applications)

Response 1: First, we would like to thank the reviewer again for the helpful suggestion regarding our choice of the 20 kHz excitation frequency. In response, we have revised the corresponding paragraph in the manuscript to clarify our reasoning—this can be found in the highlighted version of the revised PDF.

As for the broader frequency analysis, we fully agree with the reviewer’s comment. Investigating the response of the alloys across a wider frequency range is indeed something we are already planning. Our next step is to use the optimum annealing conditions identified for each material and evaluate their magnetic signal behavior over a broad frequency spectrum.

In a previous study of ours:

Samourgkanidis, G.; Kouzoudis, D. Characterization of magnetoelastic ribbons as vibration sensors based on the measured natural frequencies of a cantilever beam. Sensors and Actuators A: Physical 2020, 301, 111711.

we extensively characterized Metglas 2826MB3 as a vibration sensor and studied its frequency response. Building on that work, our current aim is to expand this investigation to other alloys such as 2605SA1 and 2714A. We see this study as laying the groundwork for that upcoming research.

We would like to point out to the reviewer that, in addition to our response to the specific comment, the highlighted PDF also includes further changes where we have added additional results and discussion. This is to ensure the reviewer is fully informed about all the updates made to the manuscript.

Reviewer 3 Report

Comments and Suggestions for Authors

The manuscript “Signal Enhancement in Magnetoelastic Ribbons Through Thermal Annealing: Evaluation of Magnetic Signal Output in Different Metglas Materials” by Samourgkanidis et al. describes the efficiency of magnetoelastic materials subjected to the thermal treatments. Despite the large number of references, they do not cover all necessary aspects of the magnetoelastic detection and do not reflect previous contributions in an objective correct way. For example, following works might be useful for discussion: C. Grimes et al. Wireless magnetoelastic resonance sensors: A critical review. Sensors 2002, 2, 294–313; Z. Fang et al. Enhancing the magnetoelectric response of Metglas/polyvinylidene fluoride laminates by exploiting the flux concentration effect. Appl. Phys. Lett. 2009, 95; Alfredo García-Arribas et al.  Sensor Applications of Soft Magnetic Materials Based on Magneto-Impedance, Magneto-Elastic Resonance and Magneto-Electricity, Sensors 2014, 14(5), 7602-7624. In any case self-citation level is very high (of the order of 25%) and it must be reduced without significant increase of the overall citing references.

Authors provide some commercial data for the ribbons indicating the level of the crystallization temperatures between 410 and 550 C. At the same time each material underwent a systematic annealing process under a range of temperatures of 50-500 C and durations (10-60 min). These data must be analyzed in a view of structural studies (TEM, XRD) as well as the magnetic hysteresis loops for each heat treatment or summary of the changes of magnetic parameters must be given. Magnetoelastic properties given without structural and detailed magnetic measurements (hysteresis loop after each heat treatment for each kind of samples) make no sense. Physical processes leading to the change of the magnetoelastic properties must be discussed in a view of the changes of structural and magnetic properties.

Apart from the magnetic properties themselves it would be interesting to get certain understanding of the thermal stability of the materials annealed at different temperatures.

Figure 2 shows the set of the ribbons prepared for annealing. What is the temperature distribution inside the furnace for the set of these dimensions? What was the temperature decay during each step of the partial removal of the ribbon samples?

Table 2 has formatting problems.

As Metglas is a trade name, it should be always written as Metglas instead of metglas.

Author Response

Dear reviewer

Thank you very much for taking the time to review our manuscript and for providing insightful and constructive feedback to help improve its quality for publication. This document contains our detailed responses to each of your comments. A revised version of the manuscript, with all modifications clearly marked and underlined, is included in the accompanying PDF file.

Comments

Point 1: Despite the large number of references, they do not cover all necessary aspects of the magnetoelastic detection and do not reflect previous contributions in an objective correct way. For example, following works might be useful for discussion.

Response 1: All three references suggested by the reviewer have been added. Additionally, some of our previous studies were removed and replaced with more relevant external references, retaining only the most essential self-citations. As a result, the self-citation ratio has now been reduced to 15%.

Point 2: Authors provide some commercial data for the ribbons indicating the level of the crystallization temperatures between 410 and 550 C. At the same time each material underwent a systematic annealing process under a range of temperatures of 50-500 C and durations (10-60 min). These data must be analyzed in a view of structural studies (TEM, XRD) as well as the magnetic hysteresis loops for each heat treatment or summary of the changes of magnetic parameters must be given. Magnetoelastic properties given without structural and detailed magnetic measurements (hysteresis loop after each heat treatment for each kind of samples) make no sense. Physical processes leading to the change of the magnetoelastic properties must be discussed in a view of the changes of structural and magnetic properties.

Response 2: The reviewer makes an insightful and valuable observation regarding the absence of structural and detailed magnetic characterization, such as TEM, XRD, or hysteresis loops following each annealing step. While these analyses would undoubtedly provide a deeper understanding of the physical mechanisms influencing the observed magnetoelastic behavior, the primary focus of this study is the practical optimization of magnetic signal output for sensing applications using commercially available Metglas alloys.

That said, we fully acknowledge the value of such structural and magnetic analyses and agree that they represent a meaningful direction for future work. Since our research group is primarily engaged in studying the sensing behavior of commercially available Metglas materials, we are always open to collaborations that would allow us to expand our methodology to include advanced characterization techniques—or even participate in the development of custom alloys tailored to specific sensing applications. We view this kind of interdisciplinary effort as essential for more comprehensive studies going forward.

Nonetheless, we believe the present study offers solid and publishable contributions by systematically evaluating how annealing influences the magnetic signal, which is directly relevant to the application focus of this research.

Point 3: Apart from the magnetic properties themselves it would be interesting to get certain understanding of the thermal stability of the materials annealed at different temperatures.

Response 3: The reviewer raises a very useful and insightful point regarding the thermal stability of the materials after annealing at different temperatures. This is indeed an important aspect that extends beyond the magnetic properties themselves and could offer a deeper understanding of the long-term performance and reliability of these alloys under varying conditions. While thermal stability was not within the primary scope of the present study, we fully agree that it would be highly valuable to explore this in future work. Assessing the structural and magnetic stability of the annealed materials over time and under thermal cycling conditions could provide important insights, particularly for sensing applications where environmental stability is critical. We appreciate the reviewer’s suggestion, which offers a meaningful direction for the next phase of our research.

Point 4: Figure 2 shows the set of the ribbons prepared for annealing. What is the temperature distribution inside the furnace for the set of these dimensions? What was the temperature decay during each step of the partial removal of the ribbon samples?

Response 4: The corresponding paragraph was revised, and additional information was included. Specifically, we used thick aluminum alloy 7075 blocks—two for each ribbon—placing the ribbon between them before inserting the entire assembly into the furnace. This setup served two purposes. First, it helped create a more stable and uniform temperature environment around the ribbon, as the metal blocks retain heat better than the ribbon alone, preventing rapid temperature fluctuations. Second, by removing the blocks with the ribbon still sandwiched between them after annealing, we aimed to achieve a more gradual cooling process. This approach avoids sudden air cooling, which could occur due to the extremely thin nature of the ribbons.

Point 5: Table 2 has formatting problems.

Response 5: We assume the reviewer is referring to the font size used for the stoichiometries in reference 45. We chose to reduce the font size of the alloy stoichiometries because, at the standard font size, they did not fit properly within the table layout and disrupted its formatting.

Point 6: As Metglas is a trade name, it should be always written as Metglas instead of metglas.

Response 6: Revised in accordance with the reviewer’s suggestion.

We would also like to inform the reviewer that, in response to suggestions from other reviewers, additional results and discussion have been incorporated into the manuscript to better support and present the objectives of this study.

Round 2

Reviewer 1 Report

Comments and Suggestions for Authors

I appreciate the author’s efforts in revising the manuscript in response to the reviewers’ comments. The current version of the manuscript has been improved significantly. However, several concerns remain unaddressed.  Prior publication, I suggest revising the manuscript to address at least the following concern:

  1. On page 9, line 232, the graph exhibits some strong linearity. The relationship between the two variables can be represented by a straight line. 
  2. On page 11, line 265, the 350-degree Celsius plot shows some strong linearity. A straight line must be applied for the graph tendency.  
  3. On page 12, line 285, the graph exhibits some strong linearity as well. 
  4. On page 7, line 300, figure 10 (b) repeats the same information of figure 10 (a). There is no need for figure 10 (b).

Author Response

Dear reviewer 

Thank you again for taking the time to review our manuscript and for providing insightful and constructive feedback to help improve its quality for publication. Here our detailed responses to each of your comments. A revised version of the manuscript, with all modifications clearly marked and underlined, is included in the accompanying PDF file. 

Comments 

Point 1: On page 9, line 232, the graph exhibits some strong linearity. The relationship between the two variables can be represented by a straight line. 

  • Response 1: Figure revised to incorporate the reviewer’s suggestion.

Point 2: On page 11, line 265, the 350-degree Celsius plot shows some strong linearity. A straight line must be applied for the graph tendency. 

  • Response 2: Figure revised to incorporate the reviewer’s suggestion.

Point 3: On page 12, line 285, the graph exhibits some strong linearity as well. 

  • Response 3: Figure revised to incorporate the reviewer’s suggestion.

Point 4: On page 7, line 300, figure 10 (b) repeats the same information of figure 10 (a). There is no need for figure 10 (b). 

  • Response 4: Figure revised to incorporate the reviewer’s suggestion.

Reviewer 3 Report

Comments and Suggestions for Authors

Submitted version of the manuscript “Signal Enhancement in Magnetoelastic Ribbons through Thermal Annealing: Evaluation of Magnetic Signal Output in Different Metglas Materials” by Georgios Samourgkanidis et al. was significantly improved in accordance with reviewer’s comments. It therefore can be accepted in the present state.

Author Response

Dear reviewer,

We sincerely appreciate your time in reviewing our manuscript and are grateful for its acceptance for publication.